# Patient Derived Chicken Egg Tumor Model (PDcE Model): Current Status and Critical Issues

**DOI:** 10.3390/cells8050440

**Published:** 2019-05-10

**Authors:** Aoi Komatsu, Kotaro Matsumoto, Tomoki Saito, Manabu Muto, Fuyuhiko Tamanoi

**Affiliations:** 1Institute for Integrated Cell-Material Sciences, Institute for Advanced Study, Kyoto University, Kyoto 606-8501, Japan; komatsu.aoi.6z@kyoto-u.ac.jp (A.K.); matsumoto.kotaro.5r@kyoto-u.ac.jp (K.M.); 2Department of Therapeutic Oncology, Graduate School of Medicine, Kyoto University, Kyoto 606-8501, Japan; tosaito@kuhp.kyoto-u.ac.jp (T.S.); mmuto@kuhp.kyoto-u.ac.jp (M.M.); 3Department of Microbiology, Immunology & Molecular Genetics, University of California, Los Angeles, CA 90032, USA

**Keywords:** CAM assay, tumor tissue, immunohistochemistry, patient tumor, H&E staining, angiogenesis, precision medicine, tailor-made drug

## Abstract

Chorioallantoic membrane assay (CAM assay) using fertilized chicken eggs has been used for the study of tumor formation, angiogenesis and metastasis. Recently, there is growing realization that this system provides a valuable assay for a patient-derived tumor model. Several reports establish that tumor samples from cancer patients can be used to reproduce tumor in the chicken egg. High transplantation efficiency has been achieved. In this review, we discuss examples of transplanting patient tumors. We then discuss critical issues that need to be addressed to pursue this line of experiments. The patient-derived chicken egg model (PDcE model) has an advantage over other models in its rapid tumor formation. This raises the possibility that the PDcE model is valuable for identifying optimum drug for each individual patient.

## 1. Introduction

Progress in genomic analysis of tumor has led to the realization that the tumors in each individual patient differ even among the same cancer types [1]. Furthermore, even the same type of tumor might differ depending on the location where the tumor arises. Thus, it is necessary to focus on each individual tumor when developing tailor-made drugs for precision medicine. Thus, patient-derived tumor models are acutely needed.

Currently, two types of patient-derived tumor models are extensively used. One type is tumor spheroids or tumor organoids, which are derived from cancer cells from patient tumor [2]. While this is a convenient model, it is not easy to recapture the actual tumor that is present in the patient sample. Various additions such as extracellular matrix components and vascular endothelial cells can be made to cancer cells to mimic the patent tumor, however, it is difficult to accurately capture tumor microenvironment. The other type of model is PDX model (patient-derived xenograft model) that is established by transplanting patient tumor to immune compromised mice [3]. However, the success rate of transplantation differs and is quite low in some cases. Furthermore, it takes time to establish a tumor on mice.

To overcome the shortcomings of the currently available models, we have been developing chicken egg tumor model as a patient-derived tumor model [4]. This model has been called the CAM assay and takes advantage of the chorioallantoic membrane of fertilized eggs. Around Day 10 of chick embryo development, the embryo is surrounded by a nutrient rich membrane. This membrane provides a place where rapid tumor growth can take place. In fact, a tumor can be formed in 3–4 days after transplanting human tumor samples or human cancer cells and close resemblance of the chicken egg tumor to patient tumor has been reported [4,5,6,7,8,9,10,11]. Because of this rapid tumor formation, one could envision a scenario where a library of 10–20 chicken eggs that represent an individual patient tumor can be established and this can be used to screen for anticancer drugs. This could lead to the development of tailor-made drugs towards an individual patient. Thus, we believe that this model, which we termed “Patient-derived chicken egg model (PDcE model)”, can contribute to precision cancer therapy.

In this review, we first describe the CAM assay, and then summarize various studies that establish patient derived tumor model in the chicken egg. We extract important lessons obtained by these studies. In addition, we discuss critical issues that need to be addressed in future studies.

## 2. The CAM Assay: Tumor Formation, Tumor Microenvironment and Angiogenesis

The CAM assay uses fertilized eggs [4,5,6,7,8,9,10,11]. When the eggs are incubated at 37 °C under humid condition (65%) and with occasional rotation, embryo is developed leading to hatching by 21 days after the fertilization. At Day 10, the chick embryo is surrounded by a nutrient rich membrane called chorioallantoic membrane (CAM) (Figure 1). The membrane provides nutrients as well as gas. Calcium from the egg shell is dissolved by an enzyme and is provided to the embryo through the CAM membrane. The CAM membrane is rich in vasculature. A window is made on the egg shell and tumor samples are placed on the CAM membrane. The tumor is formed three days after the transplantation in the case of human ovarian cancer cells OVCAR-8. When the cells are engineered to express GFP, we can obtain green fluorescent tumor, as shown in the analysis of thin section of the tumor. The rapid growth of tumor can be ascribed to the rich nutrient environment of the CAM membrane as well as to incomplete establishment of the immune system at this stage of development.

Various types of human cancer cell lines re used to form tumor in the chicken egg. According to the report by Durupt et al. [10], melanoma cells Tw12, Mel2A, M22-MEL43 or MelJuSo can be used to form tumor in chicken egg and H&E staining of the thin section from the tumor closely resembles melanoma patient tumor, as dense tumor tissue is organized in tumor nests. Similarly, colorectal cell line Lovo can be used to form tumor in chicken eggs and the tumor shows glandular structure with tubular crypts that resemble patient tumor. Furthermore, glioblastoma cell lines U87-MG or U118 can be used to form tumor that contain diffuse pleiomorphic infiltrate of fibrillary and stellate cells.

We used ovarian cancer cell line OVCAR8 and lung cancer cell line A549 to form tumor in chicken eggs (Figure 2). The tumor formed was collected and thin sections were stained with H&E, trichrome and with anti-Vimentin antibody. Similar experiments h carried out using lung cancer cell lines A549. Analysis of the tumor formed in the chicken egg by trichrome staining showed the presence of collagen and extracellular matrix [4]. Staining with antibody against vimentin revealed the presence of stromal cells. Since tumor blood vessels were found to be infiltrating into the tumor, tumor microenvironment was established in the chicken egg tumor. It is interesting that transplanting tumor cells results in the formation of tumor that consists of cancer cells, extracellular matrix, stromal cells and tumor vasculature. It appears that the tumor cells secrete factors that induce recruitment of various components to form a tumor on the CAM.

Angiogenesis plays important role for tumor growth. The angiogenesis is likely induced by angiogenetic signals such as VEGF (vascular endothelial growth factor) and EGF (epidermal growth factor), which are secreted by transplanted cancer cells. Ribatti et al. [12] investigated the inhibition of neuroblastoma-induced angiogenesis by using Fenretinide (HRP) that belongs to a class of natural or synthetic compounds structurally related to vitamin A. HRP produced anti-angiogenic effect on CAM membrane which had grafted neuroblastoma biopsy samples. Marzullo et al. [13] showed angiogenesis in hepatocellular carcinoma on CAM using human hepatocellular carcinoma that were grafted on the CAM. The neo-blood vessels of grafted tumor on CAM were counted and found to be significantly higher than in control. The angiogenic response induced by pathological implants involved basic fibroblast growth factor. This study supports the role of blood vessel proliferation by angiogenic cytokines from tumor cells, by CAM extracellular matrix and by the perivascular mononuclear cells.

## 3. Patient Derived Tumor Model Using the CAM Assay: Early Studies

Use of chicken egg model for reproducing tumor dates back to 1911 when Murphy and Rous reported that direct transmission of chicken sarcoma to developing chicken embryo can be accomplished [14]. In this study, the tumor was reduced to a pulp by grinding or by passing through a sieve. A portion of this was transplanted onto the CAM membrane and the growth of tumor was examined by histological analyses. Detailed characterization of tumor formation in the chick embryo including histological appearance, growth of tumor was reported by Karnofsky et al. [15].

Transplantation of human tumor was reported by Dagg et al. [16].They used metastatic human tumor to investigate transplantation efficiency and metastatic potential in the chicken egg. In another study, Sommers et al. [17] used a variety of human tumor samples to examine transplantation into the chicken egg. Fifty-nine different human cancers were tested with successful persistence or growth in 28 of the cases, which amounts to 47%. There was significant difference in transplantation efficiency among different types of cancer. Sarcomas were transplanted easily.

In 1991, Shoin et al. [18] reported transplantation of surgical specimens onto the CAM membrane. The specimens were freed of necrotic parts and minced with scissors and transplanted onto the CAM membrane. Growth of tumor was confirmed three days after the transplantation. The tumors tested include glioblastoma, astrocytoma, oligodendroglioma, medulloblastoma, ependymoma, malignant lymphoma and meningioma. All tumors grew while there were differences in growth rate. Sensitivity to anticancer drugs ACNU [1-(4-amino-2-methyl-5-pyrimiiyl)-methyl-3-(2-chloroethyl)-3-nirosourea hydrochloride] and MCNU was tested and a high degree of positive association between the CAM assay and the clinical outcome was noted.

## 4. Patient Derived Tumor Model Using the CAM Assay: Recent Reports

Vu et al. [4] used post operation tumor sample of ovarian cancer patient. Surgical sample of ovarian tumor was minced and transplanted onto the CAM membrane. As shown in Figure 2B, tumor was formed by Day 4 after transplantation and grew larger by Day 6. Four experiments out of five were successful in the tumor formation. H&E staining of the CAM tumor showed that the staining resembled that of patient tumor.

Among cerebral astrocytic gliomas, glioblastoma is the most malignant. Patient-derived chicken egg model was established by using glioblastoma tumors taken immediately from operating room and transplanting onto the CAM membrane [19]. The samples used were taken from 10 patients. The transplanted tumors survived up to six days. Transplanted glioblastomas exhibited necrosis, endothelium proliferation, and cellular polymorphism, which are similar to those seen with original glioblastomas. Glial fibrillary acidic protein (GFAP), vimentin, Ki67, S100 protein, neurofilament immunoreactivity were detected in the tumor produced on the CAM and infiltration of macrophages (CD68) and T cells (CD3+, CD8+) was observed.

CAM assay was established with nasopharyngeal carcinoma (NPC), a highly invasive and metastatic head and neck cancer [20]. With 35 primary tumor biopsy tissue, micro-tumor formation was observed three days after transplantation in all cases. The NPC micro-tumors induced angiogenesis and grew on CAMs with extracellular matrix interaction. This model mimics clinical features of tumor growth.

Recurrent respiratory papilloma CAM model was established by using fresh tissue samples from 13 patients [21]. Detection of cytokeratins and endothelial cells as well as proliferation evaluation were carried out by using immunohistochemical analysis against 34βE12, Ki-67, MMP-9, PCNA, and *Sambucus nigra* staining. Success of transplantation was 73% and the tumor made retained morphologic characteristics and proliferative capacity of the original tumor. Thickening of CAM and the chorionic epithelium as well as increase in the number of blood vessels in the CAM was observed.

A CAM model of laryngeal squamous cell carcinoma (LSCC) was reported [22]. Fresh LSCC tissue samples from six patients were implanted onto 15 chick embryo CAMs. Evaluation of morphological and angiogenic changes in the CAM and chorionic epithelium were carried out up to four days after the tumor implantation. Survival of the implanted LSCC tissue samples on the CAM occurred in all experiments and the tumor formed retained essential morphologic characteristics and proliferative capacity of the original tumor.

A CAM model for renal carcinoma was established [23]. This study used the clear cell subtype of renal carcinoma (CCRCC) that is highly vascularized and aggressive. Fresh tumor samples after surgical resection were transplanted onto the CAM membrane. Induction of angiogenic process was observed. Further insights into the transcriptional regulation of the model was obtained by performing a differential analysis of tumor-derived and stroma-derived transcripts.

A CAM assay for sarcoma patient tumor was established using metabolically active tumor tissue from 28 patients with bone or soft-tissue tumors [24]. Retention of essential features and immunohistochemical characteristics of the original tumors was observed. The authors note that, while longer follow-up and increased number of patients is needed to arrive at conclusion on the relationship between tumor graft behavior and natural history, the CAM assay is a potential prognostic and predictive preclinical xenograft model for tumors that are difficult to culture in vitro such as sarcomas.

Giant cell tumor of bone (GCT) is difficult to study because of the lack of suitable in vivo models. Fresh tumor tissues from 10 patients were obtained and homogenized and the suspension was grafted onto the CAM membrane [25]. All 10 samples formed solid tumors on the CAM and the tumor showed rich vascularization. Typical components of GCT such as (CD51+/CD68+) multinucleated giant cells were retained. Low proliferation rate was noted based on Ki67 staining. The tumors were composed of human cells interspersed with chick-derived capillaries.

## 5. Critical Issues Regarding the PDcE Model

A number of studies used minced tumor samples for the transplantation. Addition of Matrigel or growth factors has sometimes been used. Future studies need to establish the standard protocol for PDcE experiments which make use of growth factors or Matrigels. Another issue is to know whether multiple passages can be carried out. It may be the case that the feature of patient tumor changes after multiple passages in chicken eggs.

Many studies showed growth of tumor by the increase of tumor weight and mass. However, this may not be an accurate description of tumor growth, as the tumor formed on the CAM membrane contains multiple types of tumor components including collagen, stromal cells and immune cells. Infiltration of cells other than cancer cells may contribute to increase of tumor mass.

To compare the CAM tumor with patient tumor, a number of studies employed histochemical analysis. H&E staining results are reported in a number of publications. Immunohistochemical analyses were also carried out. In general, resemblance between the CAM tumor and patient tumor was observed. However, there are differences such as observation of much looser structure in the CAM tumor. Clearly, more detailed characterization is warranted. Detailed comparison of the CAM tumor with the patient tumor using antibodies against human proteins present in patient tumors should be carried out. Furthermore, genomic methods could be employed to evaluate the resemblance between the CAM tumor and patient tumor. Whole exome analysis can be carried out that may uncover whether genomic features of the patient tumor is retained in the CAM tumor.

## 6. Anticancer Treatment of CAM Tumor

A number of studies investigated the potential of the CAM assay for assessing efficacy of anticancer drugs. Skowron et al. [26] established the CAM assay for urothelial carcinoma (UC) by transplanting human UC cell lines. The CAM tumors were treated with cisplatin alone or in combination with histone deacetylase inhibitors (HDACi). Significant decrease in weight and growth in a dose dependent manner was observed. Vu et al. [4] used ovarian tumor specimen to form CAM model. Intravenous injection of cisplatin or doxorubicin led to almost complete elimination of the tumor in two to three days. Potential side effect of anticancer drugs can also be assessed using the PDcE model. In the work by Vu et al. [4], various organs were cut out and their size, color and shape were examined. This may allow estimation of a minimum dose that inhibits tumor but does not cause organ damage.

Efficacy of antiangiogenic drugs has been evaluated using the CAM assay. Marimpietri et al. [27] examined antiangiogenic effect of the combination of vinblastine and rapamycin using neuroblastoma CAM assay. Ozcetin et al. [28] examined anti-angiogenic effects of imatinib using CAM assay established by transplanting urothelial carcinoma (MB49) cell suspension.

Photodynamic therapy of malignant ovarian tumors on CAM was evaluated by Ismail et al. [29]. In this experiment, methylene blue was used as a photosensitizer and was administered by using a liposome formulation. Light irradiation was carried out by using a krypton laser. Two days after PDT, significant decrease of the tumor was observed. Radiosensitizing activity of etanidazole, a hypoxic cell radiosensitizer, was examined by Abe et al. [30] using the CAM assay established by transplanting mouse mammary cells. After establishing the tumor, etanidazole was intravenously injected and the egg was exposed to X-ray at a dose of 4 Gy/min using a 150 kVp X-ray generator. Significant effect on tumor growth was observed. The CAM assay can also be used to assess oncolytic viruses. Durupt et al. [31] used melanoma tumors on the CAM to test adenoviral vector carrying the gene encoding the fusion protein of parainfluenza virus type 5.

## 7. Potential of PDcE Model for Identifying Tailor-Made Drugs

The CAM assay provides a simple and versatile model to establish patient-derived tumor model. Various tumor models have been successfully developed. The tumor formed on the CAM membrane can be treated with anticancer drugs and other methods. Thus, one could envision a scenario depicted in Figure 3. A tumor from an individual patient will be used to establish tumor on the CAM membrane. This tumor can be passaged to build up a library of eggs that represent a tumor from an individual patient. This library can be used to test anticancer drugs resulting in the identification of optimum drugs. This could lead to the identification of tailor-made drugs for each individual patient. Further investigation is needed to examine whether this scheme can be realized in the future.

## Figures and Tables

**Figure 1 cells-08-00440-f001:**
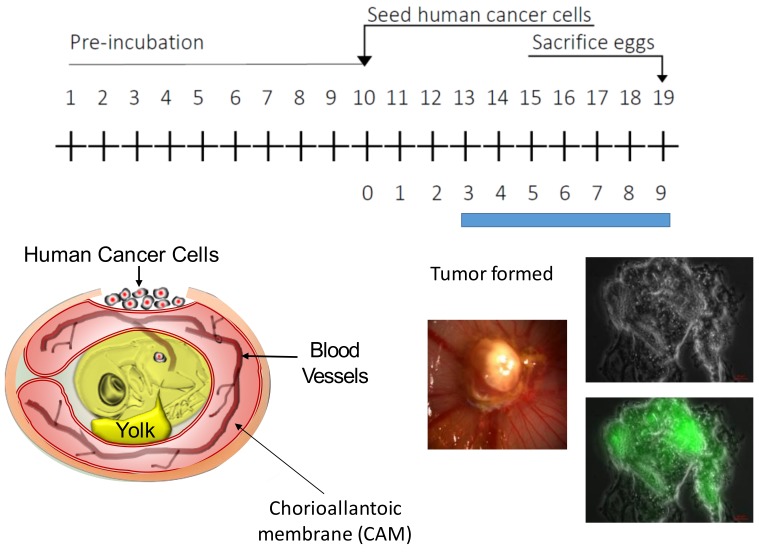
The CAM assay. Fertilized eggs are incubated at 37 °C and under 65% humidity with occasional movement. By Day 5, embryo is surrounded by a nutrient rich membrane called chorioallantoic membrane (CAM). A window is made on the egg shell and human ovarian cancer cells OVCAR8 are placed on the CAM membrane. Three days after the transplantation, a tumor is formed. When GFP expressing cancer cells are used, we can observe green fluorescent tumor. Modified from Figure 1 of Vu et al. [4].

**Figure 2 cells-08-00440-f002:**
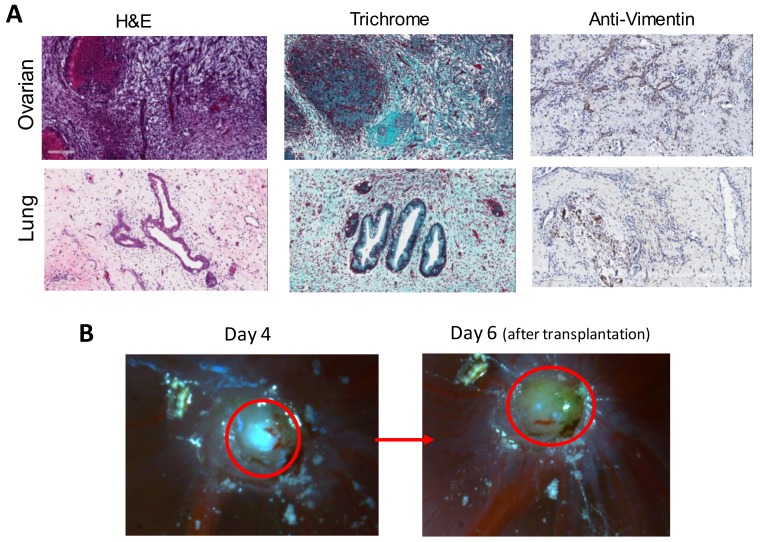
(**A**) H&E, trichrome and anti-vimentin staining of the thin section of CAM tumors formed by transplanting ovarian cancer cells OVCAR8 and lung cancer cells A549; and (**B**) ovarian cancer patient tumor was minced and transplanted onto the CAM membrane. Tumor growth was followed. Modified from Figure 2B and Figure 8A Vu et al. [4].

**Figure 3 cells-08-00440-f003:**
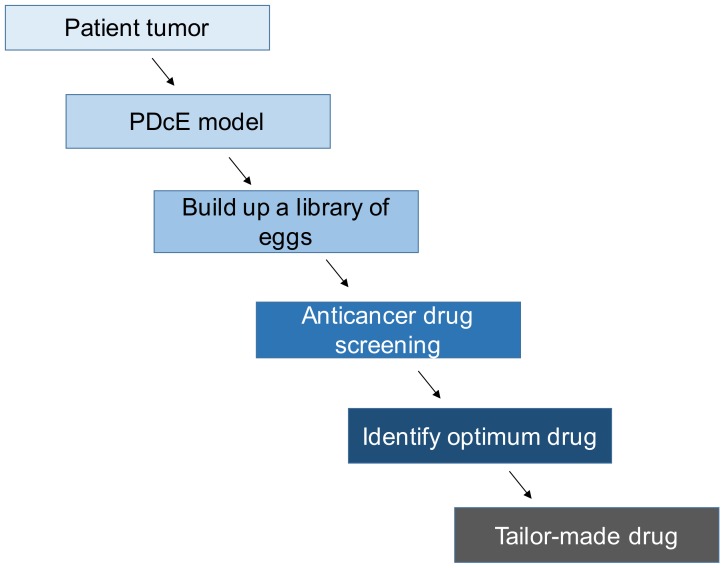
A scheme for the use of PDcE model to develop tailor-made drug for an individual patient.

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
