# Peer review of "Patient Derived Chicken Egg Tumor Model (PDcE Model): Current Status and Critical Issues"

_cells, 2019, doi:10.3390/cells8050440_

Round 1

Reviewer 1 Report

The manuscript presented by Komatsu et al reviewed recent articles of CAM assay on tumors, especially findings from patient derived tumors, which could be of interest to the related researchers. CAM assays have long been recognized as a method to study angiogenesis, tumor cell invasion and metastasis. In the present review, the author focused on its usage for patient specific tumor formation, making it another PDX method besides nude mouse transplantation and organoids. The manuscript is concise and well written. The information is well presented and described and thus I recommend publication in Cells.

Author Response

We would like to thank the reviewer for positive comments. We did check minor spell mistakes and corrected them.

Reviewer 2 Report

This review is timely and will be of interest to readers of Cells, MDPI. The CAM assay using fertilized chicken eggs lends itself to patient derived models and can become very useful for personalized cancer treatments.

Commenst

line 64:  is it 3 days for all tumor types?

line 97: what other components are recruited to the tumor site?  are they chicken cells?  please elaborate

Author Response

Thank you for the review. In response, we made the following changes to the manuscript.

1. The reviewer asked whether the tumor formation in the CAM assay is 3 days for all tumor types. Our experience with a variety of tumor types is that the tumor formation occurs in 3-4 days in general. However, there are some differences between tumor types. Therefore, we changed the sentence to read, "Tumor is formed 3 days after the transplantation in the case of human ovarian cancer cells OVCAR-8" (Line 64). 

2. Line 97: This sentence was misleading. Therefore, we corrected the sentence to the following. "It appears that the tumor cells secrete factors that induce recruitment of various components to form a tumor on the CAM".

Reviewer 3 Report

Review deal with  chicken egg tumor model as a patient-derived tumor model as this model has been called the CAM assay and takes advantage of the chorioallantoic membrane of fertilized eggs.

1. My question to you will be any molecular characterization of the PDcE model has been done? If there are studies dealing with this validation please mention in discussion.

2. Repharase:? lines 180-183:

A number of studies used minced tumor samples for the transplantation. Addition of Matrigel 180 or growth factors has sometimes been used. Further study is needed to understand what the optimum 181 protocol for treating tumor samples is. A systematic study to examine effect of the addition of growth 182 factors or Matrigel needs to be carried out. 

TO:

A number of studies used minced tumor samples for the transplantation. Addition of Matrigel 180 or growth factors has sometimes been used. Future studies need to establish the standard protocol for PDcE experiments which make use of growth factors or Matrigels.

Author Response

Thank you for the review.

Yes, there are some molecular characterization carried out. In fact, we included this in section 4 where we discuss recurrent respiratory papilloma CAM model. In section 5, our description gave the impression that only H&E staining was carried out. We added a sentence about immunohistochemistry analyses and corrected this point. (Lines 194-197)

We rephrased the sentences concerning future studies according to the suggestion by the reviewer. (Lines 184-186)